# Value of stakeholder engagement in improving newborn care in Kenya: a qualitative description of perspectives and lessons learned

Jacinta Nzinga ,[1] Caroline Jones,[2,3] David Gathara,[2] Mike English[2,3]

¹Health Services Unit, KEMRI-Wellcome Trust Research Programme Nairobi, Nairobi, Kenya
²Health Systems Research and Ethics, KEMRI-Wellcome Trust Research Programme, Kilifi, Kenya
³Nuffield Department of Medicine, Oxford University, Oxford, UK

**Correspondence to**
Dr Jacinta Nzinga;
jnzinga@kemri-wellcome.org

## ABSTRACT

**Objective** Embedding researchers within health systems results in more socially relevant research and more effective uptake of evidence into policy and practice. However, the practice of embedded health service research remains poorly understood. We explored and assessed the development of embedded participatory approaches to health service research by a health research team in Kenya highlighting the different ways multiple stakeholders were engaged in a neonatal research study.

**Methods** We conducted semistructured qualitative interviews with key stakeholders. Data were analysed thematically using both inductive and deductive approaches.

**Setting** Over recent years, the Health Services Unit within the Kenya Medical Research Institute (KEMRI)-Wellcome Trust Research Programme in Nairobi Kenya, has been working closely with organisations and technical stakeholders including, but not limited to, medical and nursing schools, frontline health workers, senior paediatricians, policymakers and county officials, in developing and conducting embedded health research. This involves researchers embedding themselves in the contexts in which they carry out their research (mainly in county hospitals, local universities and other training institutions), creating and sustaining social networks. Researchers collaboratively worked with stakeholders to identify clinical, operational and behavioural issues related to routine service delivery, formulating and exploring research questions to bring change in practice

**Participants** We purposively selected 14 relevant stakeholders spanning policy, training institutions, healthcare workers, regulatory councils and professional associations.

**Results** The value of embeddedness is highlighted through the description of a recently completed project, Health Services that Deliver for Newborns (HSD-N). We describe how the HSD-N research process contributed to and further strengthened a collaborative research platform and illustrating this project's role in identifying and generating ideas about how to tackle health service delivery problems

**Conclusions** We conclude with a discussion about the experiences, challenges and lessons learned regarding engaging stakeholders in the coproduction of research.

### Strengths and limitations of this study

► Strengths from this article include emphasis on involvement; understanding who is and should be involved, when should this engagement occur (ie, at what points in the research process) and how this engagement should be done (ie, what are the approaches to engagement that yield the results).

► Furthermore, successful participatory processes require; openness of dialogue with a genuine empathy for others' perspectives; active listening and courtesy; early and ongoing voice and creating meaningful decision space throughout the engagement process.

► However, the limitations of this study include complications by a number of context and resource-based factors including competing priorities, tension among stakeholder groups, high staff turnover and lack of commitment.

► There is a need for more empiric work to develop and apply explanatory theories, frameworks and models to better understand how participation occurs, under what contextual settings and what is produced.

## INTRODUCTION

Recent literature has underscored the value of health policy and systems research (HPSR) as an intervention for systems strengthening.[1] In the last decade, there has been an increased demand for embedded health systems research in low and middle-income countries (LMICs), as leverage for more socially relevant and responsive research and for more effective uptake of evidence into action/policy/practice.[2 3] Furthermore, implementation research has highlighted the need for context-specific research evidence as part of solutions to address the translation of knowledge into practice.[4–6] However, the uptake of research findings heavily depends on the credibility of the information produced, which is in turn dependent on trusted local stakeholders' expertise and their active, meaningful involvement throughout the research process.[7–9]

This paper provides a brief description of our (a health research group) history of more than 15 years of engaging with stakeholders and conducting health services' research in Kenyan hospitals and explores the relational and organisational processes underlying network activities; examining the spaces in which stakeholder engagement occurred over a number of years during work that focused on hospital improvement.[10–12] It then provides a critical analysis of the most recent lessons learnt through a description of a study aimed at understanding how local structural, contextual and cultural factors influenced the research–policy–practice engagement process in a recently completed health systems research project. The aim is to provide a better understanding of the requirements of embedded participation in responding to local problems.

## Study background

The Health Services Unit (HSU) of the Kenya Medical Research I-Wellcome Trust Research Programme started working closely with the Ministry of Health (MoH) of Kenya in 2004 developing and implementing research on facility-based care to improve child and newborn survival.[13–15] Early work focused on developing and implementing a multifaceted intervention aimed at improving paediatric inpatient care in district hospitals in Kenya.[16] Data collection included long-term participant observation and continuous reflection on the positionality of study team members embedded in the study hospitals.[17 18] To allow engagement with stakeholders, regular evidence synthesis meetings and feedback meetings were held with the hospitals. There were bi-monthly phone calls to understand how the intervention was unfolding as well as formal and informal discussions and consultations with the stakeholders to understand their interest in the engagement. A key lesson from the project was that changing practice and system hospitals required specific collaboration with partners who are usually considered the subjects of research.

Consequently, driven by the need for systemwide improvement, the HSU partnered with the MoH, the Kenyan Paediatric Association and 14 county (district)-level hospitals in 2010 to create a clinical information network (CIN) spread over 16 counties in eastern, western and central Kenya.[19] The network aimed to produce high-quality process and outcome data from individual admissions to paediatric wards in Kenyan hospitals and use these data to inform improvement strategies. Through collaborative working, the network has grown into a community of practice aimed at slowly changing hospital culture through sustained engagement, peer support and linking hospitals within the network.[20] The effects of the CIN platform, critically explored through formative explanation and theory of change, are documented elsewhere.[21]

Through these projects, the research team began to learn from stakeholders how contexts shape service delivery and how relationships between the research team,

health managers and health workers develop and shape the delivery of the interventions over time.[22 23] However, this research process involved limited true coproduction, partly because research funding provided limited support for extensive work of this kind. Furthermore, it was apparent that the practice of embedded HPSR in LMICs was, at that time, not very well defined and that trial-and-error strategies like our own were often applied.

Over time, the research group developed a more deliberate and collaborative approach that was taken forward in subsequent projects including the HSD-N project detailed below.

### The HSD-N project: 2013–2018

As a research team, concerned by the high neonatal mortality in Nairobi, we held consultative meetings with the County Government of Nairobi and other key stakeholders. Together, and while drawing on our 10 years' research experience on quality of care,[24–26] we codeveloped the HSD-N project with key stakeholders. The project aimed to address the challenges influencing the delivery of essential inpatient newborn services in Nairobi County with a particular focus on nursing care, which was highlighted by all stakeholders as a neglected topic (figure 1).

The initial approach to conceptualising how gaps might be addressed was informed by Kenyan policy objectives, specifically the focus at national policy level on task shifting[27] and early discussions with the Nairobi City Council in which concerns over how newborn care was delivered across the public, private and faith-based sectors were raised. In light of the prevailing policy environment, our research included an explicit aim to explore the potential of task shifting through the use of health-care assistants to support nursing care as one potentially important component for improved newborn care practice in Kenyan and possibly other LMICs.[28 29]

The HSD-N project took place in three phases (figure 1). At the heart of this work, it was a strategic approach to researching and intervening in the health system based on collaborative engagement from the outset. Building on relationships developed from previous projects we began to forge new linkages with powerful (had authority to influence key policy decisions in newborn care) professionals including regulators, health professional bodies, private institutions and other major decisions-makers in health in Kenya.[30] This stakeholder network was a core facilitator for truly collaborative and coproduced research.

### Phase 1 (2014–2015)

The existing links developed by the HSU over the years allowed an initial drafting of a list of key stakeholders likely to play a critical role in the conduct and impact of research addressing nursing service policy and practice issues.[31 32] The list was collaboratively reviewed by the research team and initial stakeholders with more stakeholders added following certain strategic considerations.

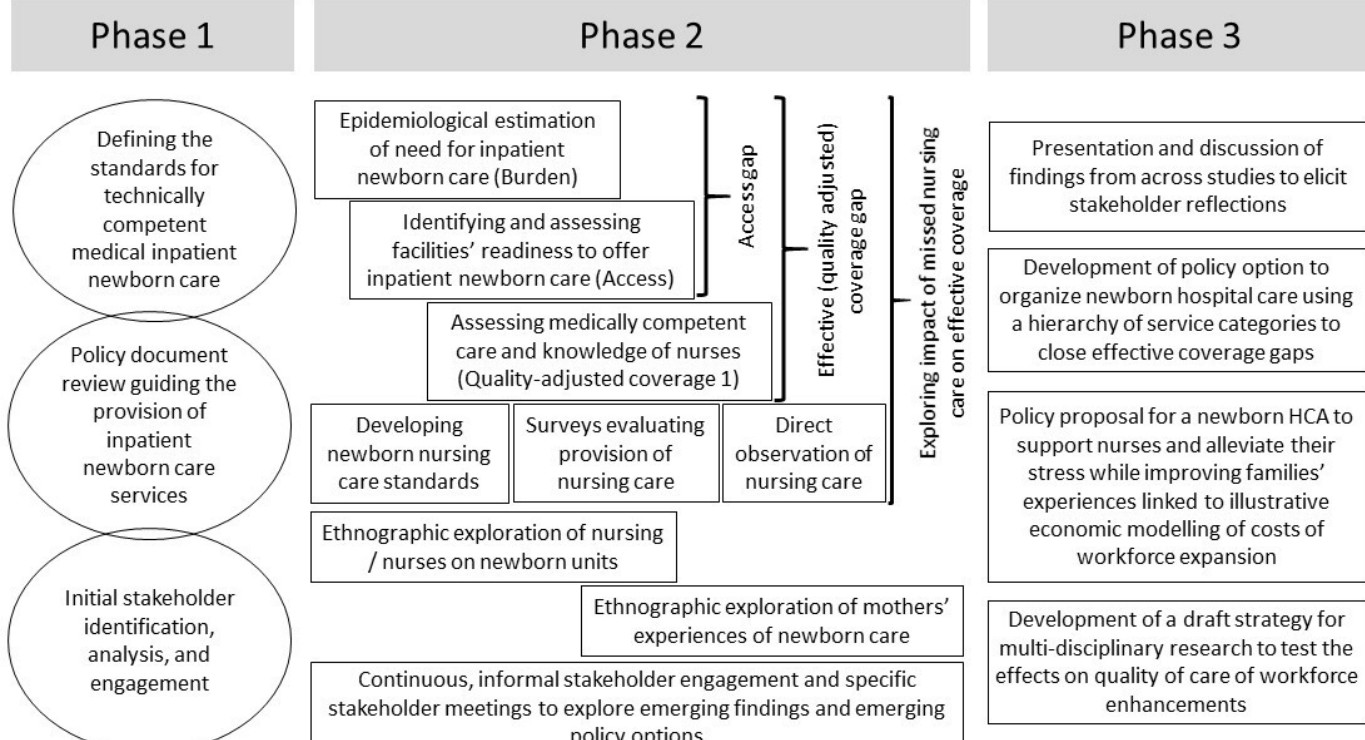

**Figure 1** Schematic of HSD-N research components, their inter-relationship and infused stakeholder engagements throughout the research cycle. HCA, healthcare assistant; HSD-N, Health Services that Deliver for Newborns.

These included the projects' core research questions; the power and interests of those who would be responsible for making decisions informed by the research and the individuals and groups that would be affected by such decisions. Specifically, during stakeholder meetings, the appropriateness and effectiveness of the research approach adopted were heavily dependent on learning from and listening to these stakeholders.

### Phase 2 (2015–2017)
The empirical data collection for the HSD-N project started with two distinct bodies of work (see Fig 1)[15 33]. During this empirical phase of the project, engagement activities included stakeholder engagement meetings and workshops, various trainings and hospital feedback meetings on empirical findings (table 1).

### Phase 3 (2017–2018)
Alongside empirical data collection, a series of stakeholder workshops with nursing and neonatal care experts helped define core standards for care of sick newborns in Kenyan hospitals.[25 34] The stakeholder workshops focused on the capacity required to provide an essential package of services for sick newborns; understanding the nursing time/skills needed for effective delivery of interventions and were complimented by hospital feedback meetings and various topic-specific meetings as shown in table 1.

To provide an in-depth understanding of how the HSD-N project was developed and implemented in practice, we present a chronological timeline of the research process and how the 'engagement platform' developed,

identifying the key engagement activities that were influential in enabling coproduction during the lifetime of the project (table 1).

## METHODS
### Research design
This was a qualitative exploratory study.

### Study setting
To explore the content and consequences of the HSD-N engagement activities over the project period, we conducted key informant interviews and preplanned observation of HSD-N meetings within Nairobi County.

### Patient and public involvement
Patients were not involved in setting the research question or the outcome measures, but key public stakeholders who were part of the HSD-N collaborative group and described in this paper were consulted in the design, conduct and dissemination of the study findings.

### Key informant interviews
To build our understanding of how neonatal care is perceived within policy and practice environments, we tracked the continuing purposeful engagement with stakeholders exploring the influence of stakeholder–researcher interactions. Six months before the end of the project, we conducted in-depth interviews with purposively selected key informants with potential policy influence, including: The Nursing Council of Kenya, National

**Table 1** Chronological representation of research engagement and contribution of the HSD-N project in shaping engagement and co-production of research

| Year | 2014 | 2015 | 2016 | 2017 | 2018 |
|---|---|---|---|---|---|
| **Engagement platforms** | | | | | |
| Meetings | Two meetings with representatives from the Nairobi County health management team, with the universities, KP and MoH. These meetings were held during the drafting of the proposal through to submission for funding | One meeting with County Executive Member for Health Services Stakeholder meetings introduction to the HSD-N project Meeting on estimating the requirement for inpatient neonatal care and neonatal burden of disease | Expert meeting on developing Neonatal Nursing Standards of Practice Stakeholder meetings on Estimating the requirement for inpatient neonatal care Basic standards of quality newborn care Results of the Nairobi newborn study on neonatal service provision | Nairobi Newborn Study feedback and presentation of report meeting Feedback meeting on results on the context issues for neonatal nursing task shifting Hospital specific feedback meetings on task sharing in practice An introduction to survey work on missed neonatal care meeting | Healthcare assistants costing meeting Cross-site hospital feedback meetings on task sharing in practice Developing nursing indicators meeting Feedback on missed care survey work meeting |
| Workshops | | Checking newborn epidemiological estimates with newborn experts Check the facilities we identified for the survey Disseminate the facility survey findings | 'Fact-check' workshop on the early facility survey findings Expert workshop meeting on developing Neonatal Nursing Standards of Practice | Two workshops on NHCA scope of practice and training, On hierarchical task analysis (two of these) On nursing missed care questionnaire design | One on levels of neonatal care One on costing. |
| Interviews | | Stakeholder mapping and collecting views on task-shifting with paediatric and nursing experts, academic stakeholders | | | End of project interviews with 14 stakeholders |
| Training | | | Hierarchical task analysis meeting | Missed care observational methods training | |
| Hospital specific feedback meetings | | All through | | | |
| Multi-disciplinary quarterly researcher reflective meetings | | All through | | | |

HSD-N, Health Services that Deliver for Newborns; KP, Kenyatta University; MoH, Ministry of Health; NHCA, neonatal healthcare assistants.

Nursing Association of Kenya, Kenya Pediatric Association, various nursing training schools, private organisations and frontline workers. Selected participants included both men and women, with varied years of working experience and with specific expertise in newborn care. Although the HSD-N project was geographically Nairobi focused, many of the groups represented national-level stakeholders.

The interviews were guided by a pilot-tested interview guide that focused on what drove individuals to be part of the stakeholder network, their understanding of the project, nature of involvement, how their inputs were gathered and any impact of their involvement. All interviews were conducted in English, within participants' work premises and lasted 40– 60 min. The interviews

were audio-recorded following informed consent from participants and field notes taken during and after the interviews.

**Data analysis**

Data were analysed both inductively (emerging from the interview data and observation notes) and deductively driven by a priori themes and coded using Nvivo Qualitative software. Data were coded around the purpose and mechanisms of engagement, researcher–stakeholder relationships and how local structural, contextual and cultural factors influenced the process of research–policy–practice engagement.[35 36] Through critical analysis of the empirical data and reflexivity, we developed a rich

**Table 2** Description of the emerging themes and subthemes

|   | Themes | Sub-themes |
|---|--------|-----------|
| 1. | Classification and description of stakeholders | Stakeholder identification process |
|   |   | Nature of engagement |
|   |   | Level of engagement |
| 2. | Context and nature of engagement | Perceived value of stakeholder meetings |
|   |   | Role of feedback in shaping engagement |
|   |   | Strategies used in managing voices of the various stakeholders |
| 3. | Interpreting the HSD-N engagement | Technical capacity to engage with various research topics |
|   |   | Ability to implement lessons from research project |
| 4. | Facilitator and barriers of the engagement | Early engagement in the project |
|   |   | Creating safe spaces for deliberations |
|   |   | Multi-level actor engagement |
|   |   | Stakeholders' competing priorities |
|   |   | Perceived 'poor' compensation |
|   |   | High stakeholder turn-over |

HSD-N, Health Services that Deliver for Newborns.

description of the concerns and interests of stakeholders likely to be affected by the research findings. The findings are summarised under four main themes: classification and description of stakeholders; interpreting the HSD-N engagement; barriers and facilitators of engagement and the context and nature of engagement.

## RESULTS

The results we present are based on interviews with 14 selected stakeholders at the end of the HSD-N project in 2018 and presented under four main themes (see table 2).

### Classification and description of stakeholders

Stakeholders of the HSD-N project were primarily from the public sector, which provides the majority of neonatal care in Nairobi.[37] However, some stakeholders from private and non-for-profit organisations were included. None of the stakeholders was compensated for their time on the project although there were in-built mechanisms to build capacity through short trainings on research and select relevant quality improvement topics. The roles of stakeholders in the HSD-N project were linked to four key project activities (table 3): (1) study planning (includes codesign of the research questions, (2) study design

procedures and development of study tools, (3) study implementation (as study participants, development of modelling scenarios or training curricula and drafting nursing standards) and (4) interpretation and translation (ambassadors of implementation and change).

> 'R: This one (HSD-N) was different thing … in the initial phases of the design of the project we were involved as part of the team that we were actually designing the tools and refining them and even having consensus. So, this was good… because I participated more.' Female senior university lecturer
> 'I collected some data, they involved me in data collection on task sharing and I felt well… I felt engaged, like I can actually give people who are here, who work in Kenyatta and get their views' Female nurse manager

To fully understand who should be engaged, when should this engagement occur (ie, at what points in the research process), we explored the nature of the various engagements and present in table 3.

### Context and nature of engagement processes

In table 3, we provided a categorisation of stakeholders, the nature of engagement and stakeholders' perceived roles in the project over the 4-year implementation period.

We also sought stakeholder's opinions as to why they think they were invited to be part of this project and why they continued engaging with the project activities. Most participants reported they believed they had important contributions to make and that the project allowed an avenue for this while others joined out of personal interest:

> 'R: Personally, I love something that is out of what I do every day… like research can help in boosting, … I can change in the unit…I love doing different things from the norm that is why I felt I can be part of this. This project is beyond relevant… because our unit is… we handle 200 babies and it is like 50% will go 50% will die. You know if are in such a project …you can do something about the situation… well I believe it is very relevant.' Male paediatrician
> 'R: Well, there is always the person part of it [HSDN] that you interact with people because quite often when we are working, everybody is just too busy to interact with each other' Female paediatrician

As mentioned above, the Health Services that Deliver for Newborns (HSDN) project ran several activities as part of stakeholder engagement using concept mapping and focus groups, and all these activities were documented and archived to inform the process and success of the project (refer to table 1 for type and purpose of the meeting). Stakeholders described these meetings as useful 'engagement spaces' that provided opportunity to not only discuss various aspects of the research but to also get updates regarding the project and included learning opportunities.

Particularly valued was provision of regular feedback, ensuring that the most knowledgeable stakeholders in the subject matter were present and that their views were sought and incorporated into the

**Table 3** Description and roles of HSD-N stakeholders

| Stakeholder categories | Policymaker | Regulator | Professional association | Training institutions | Health managers | Health workers | Researchers |
|---|---|---|---|---|---|---|---|
| | Department of Monitoring and Evaluation Department of Nursing services, Ministry of Health (MOH) WHO United Nations International Children's Fund | Nursing council of Kenya (NCK) | Kenya Paediatric Association (KPA) The National Nursing Association of Kenya (NNAK) | Kenya Medical Training College University of Nairobi AgaKhan University Hospital (AKUH) Kenyatta University | Ward and departmental managers of; Public hospitals Mission hospitals Private hospitals | Nurses, medical officers and clinical officers of; Public hospitals Mission hospitals Private hospitals | Multi-disciplinary team of researchers from Kenya Medical Research Institute-Wellcome Trust Research Programme, AKUH, Strathmore University Oxford University Warwick University |
| **Nature of engagement** | | | | | | | |
| Consultative | Collaborated with the team in study design, implementation. Advised on the political and regulatory landscape | Collaborated with the team by offering advice on study implementation. Advised on the political and regulatory landscape | Advised on the political and regulatory landscape | Provided technical theoretical and practical advice during various sessions of evidence generation Major voice in design of neonatal health care assistants (NHCA) scope of work and preliminary curriculum plus potential salary | Provided technical advice during various sessions of evidence generation Significant voice in shaping NHCA roles (some were already using helpers informally or in private sector more formally) and also suggestions on the political presentation of the NHCA cadre Useful reflections on the practical realities in routine service provision | Provided technical advice during various sessions of evidence generation and reflective of the practical realities in routine service provision | |
| Involved | | | Involved in aspects of study implementation, including data collection Offered expert critique and suggestions on improving emerging findings (eg, neonatal burden estimation) | | | Involved in aspects of study implementation, including data collection | Mainly involved in evidence generation, incorporating the technical advice of various stakeholders in the analysis Collating the interpretation of findings and implications on policy and practice |
| **Interpretation and translation** | | | | | | | |
| Strategic endorsement | Added credibility to the research evidence and enabled other big players to be part of the deliberations (eg, NNAK, NCK) Statutory agreement of translating study findings into policy recommendations | Added credibility to the research evidence and enabled other big players to be part of the deliberations (eg, NNAK, NCK) Offered reflections on feasibility of translating evidence into practice | Acted as ambassadors of change and implementation of study findings | | | | |

HSD-N, Health Services that Deliver for Newborns.

final reports. Feedback meetings allowed researchers to check understanding and modify interpretations and key messages. In particular, efforts by the research team to understand why there may be support or resistance to some of the potential recommendations was also important.

However, during these meetings, it was not always easy managing differing views and reactions regarding emerging recommendations, and it was particularly challenging dealing with the varied power dynamics from different groups and individuals. However, we observed stakeholders' free and frank exchanges in voicing opinions, open disagreement and, on occasions, the research team taking on arbitration roles to ensure all voices were heard. During interviews, stakeholders recounted the various strategies they drew on in making sure they were heard and in respectfully disagreeing with opinions as illustrated below. The nature of engagement that emerged was mainly both consultative and collaborative, which enabled the cumulation of understanding and development of meaningful relationships.

### Interpreting the HSD-N engagement
We were interested in the stakeholders' articulation of how research findings were established and their influence over such findings as this would potentially benefit effective implementation.

'R: In the meetings there are those people who participated in the research projects and also in the meetings, so it gave the project authority. and it made sense to the people who participated. When we hear that those who participated are also here, we also appreciate that report and the feedback and the evidence that is being presented.' Male, Professional association
'I think was a very exciting journey because we were able to share with each other, with the paediatric association, to discuss with the paediatricians and even have the consensus of where we need to be. I also I think the other exciting journey came in when I was involved as part of the cohort to do the publication.' Female, Regulatory Council
'R: If they are not listening then you still continue shouting there is no other language but of course occasionally you have to sit down think of another strategy. In such a situation that is the time when you think of who else has a voice, you have to think of who else could be having the same mind as mine so that you put the two voices together and we see whether we can be heard that is one strategy.' Female frontline nurse

During the interviews, we reflected with stakeholders about (1) their technical capacity and ability to engage with the varied research topics, (2) how their feedback was incorporated into the project and (3) ability to implement lessons from the project. Examples are provided below:

On ability to conceptually engage with the research, with experiential understanding of the research problem, stakeholder reported the importance of having technical capacity to engage and also felt

that their feedback influenced the research process. Furthermore, stakeholders, who had the ability, described application of new clinical information in their hospitals.

'I also participated in the review of the procedure manual so I knew the procedures and when you tell me that a nurse assistant will be able to give fluids or to do blood transfusion then am going back to the rationale of that procedure' Female nurse manager
'Just the voice, convincing people that it is worth taking it up, and the fact that I am a trainer… I understand all curriculum and I understand the needs in the service delivery units I think with that in mind it [engagement] has enabled me to work with whoever towards achieving the goals of the project.' Female lecturer, training college
'R: Every time we came out of the meetings we would also come and improve things within the facility. So, there is already been a positive feedback and in fact use of the learning that we have done within the facilities.' Female Paediatrician

According to the stakeholders, the process of cultivating long-term researcher–stakeholder relationships meant respecting each other's time and commitment, continuously reviving interest in the project and clearly communicating and negotiating expectations.

### Barriers and facilitators of the HSD-N engagement process
We learnt to be sensitive to stakeholders' time commitments as this was perceived as highly important for continued engagement. Understanding how stakeholder integrate on-going research activities into daily work enabled bringing together people from various levels of the health sector building multilayered perspectives of the research project in terms of its implementation.

As a research team, we learnt that successful stakeholder engagement required early involvement in project design, providing prereadings to enable informed discussion, creatively using 'icebreakers', especially when engaging stakeholders with differing experiences/perspectives and clearly communicating the anticipated commitment of time and level of engagement.

'R: That [stakeholder engagement] kind of interaction has been quite good. Quite often when the team sent out mail, some of us try to say okay 'I have been sent this and I think I need to meet my obligation'. That communication I think it has been quite good. And top of that, it hasn't been overwhelming because for this project we have been given adequate time to be able to address things and of course most of those documents they have been sending have not been these huge heavy documents that bog one down' Female lecturer, training institution

Despite the positive feedback, the engagement over time also had some limitations. The most commonly reported barriers not only included competing priorities by most of the stakeholders and, therefore, a struggle to find time for the meetings but also, perhaps paradoxically, limited time allocated for deliberations during the

'R: The meetings were fairly regular and fairly spaced …so would have like once in six months, so I think the regularity was good because most people are really pressed on time' Female, frontline nurse facilities, that is terms of the levels public, private and then we have lecturers, we have doctors and the Nursing Council. I think it's a good way

'R: I realized we are meeting with a variety of stakeholders, from different because they are able to listen to us the people on the lower level. What we are going through…, they were able really to compare and see actually this is something that will work." Female, Professional Association

"R: The study reports are available for most of us… we are able to go through the whole process of the study we are able to go through and it is available, so I think that is also a strong area for the study group.' Male, training institution

stakeholder meetings. Finally, sometimes the difficulty in finding the appropriate representation of stakeholders that the project sought to engage was a challenge. In other instances, the problem was the issue of sending a different representative of a group or organisation to the meetings each time. Often new people struggled to understand the project's background, progress and future aims. Similarly, poor representation of administrative/managerial groups especially from the county which has high staff turnover diminished interest, commitment and ability to follow research activities was perceived by stakeholders as a threat to utility and sustainability

'R: I can say time…time factor has been… cause most of the time am not usually released from here [hospital x] I try to create my own time, so if you say like am here for the whole day, that means I have to squeeze in 2 shifts, because I usually report here at around 7:30am to 5:30pm so those are 2 shifts, I need to get 2 people to cover my shift but I really don't mind…I really don't mind.'
R: Yes, you know sometimes we just want to go to another place.
M: That is not our office?
R: Exactly, if we can be able to see how resources can be able to work for a two day out of the town. So, my issue is I never even participate fully…I am always called to work, so I have to keep rushing. So, I thought at sometimes that if allowable we could actually get out of your offices and we work even though it is one day we actually work until whatever time even if it is midnight. That way I feel it would be more relaxed. I felt that it was a bit tensed and like we need to make this decision, and this is the period we have, and we have to hurry up. I was okay with that speed, but I think at some level maybe we were leaving some other people dragging behind, so could we allocate a bit of time and also out of town. Female Lecturer, Training institution

'R: The things that were less exciting is that the administration aspect of the project involvement was missing. When I noted that the in charges of the unit or the hospitals were missing in this study, to me I felt your likelihood of sustainability of the good things you have done is questionable and likely to have a challenge. …because there was no commitment from the administration.' Male paediatrician

## DISCUSSION AND CONCLUSION

Our findings highlight the importance of purposefully selecting stakeholders to fit project needs. Clearly defining roles and expectations for both researchers

and the stakeholders and providing continuous feedback appeared key drivers of meaningful and impactful engagement.[38 39] Perhaps more vital is mapping the dynamic nature of stakeholder's involvement over a projects' lifetime and creating opportunities to share ideas and views in 'safe' settings. We emphasise the importance of involving across-system actors who are often overlooked in such processes, for example, from frontline health workers who may help articulate and validate the research priorities and as implementors of recommendations to policymakers and regulators with the authority to formalise recommended practices.

We have shown that embedded participation requires investing in social capacity in form of openness of dialogue active listening and courtesy and respectful consideration of ideas contributed. When all elements are present, then participation processes are likely to increase involvement and legitimacy, and if participants feel that their views are valued and used, this ultimately enhances how the research may be used in decision-making. However, as we learnt, participatory processes are complicated by a number of context and structural issues including managing divergent opinions, tensions and mistrust, which require interpersonal and facilitation skills, which not all academics are trained in or endowed with.[40]

Furthermore, there also needs to be more reflection on how to meaningfully measure the worth of embedded participation.[41 42] This involves including both outcome and process factors and acknowledging that participatory processes typically require long time frames to build awareness and work through existing stakeholder dynamics.[43 44] There ought to be open discussions on how embedded engagement influences research processes; the significant risks for academics, who are required to adopt practices far from those traditionally taught and having to continuously manage group dynamics. There is need for reviewing funding structures in lieu of conflict between the emergent, dynamic yet invaluable role of engaging stakeholders in research versus strict timelines tied into specified deliverables. Finally, the need for clearly defined methods for evaluating participation, including focus on power analysis and more studies on developing and applying explanatory theories that better articulate how participation occurs within the relational contexts of coproduction.

**Acknowledgements** The HSD-N research team, particularly Elizabeth Kyala who helped with archiving the stakeholder engagements and the rest of the HSD-N Collaborative Group who made this work possible. We are also grateful to the health workers, and colleagues representing various stakeholder institutions who made this work possible. This paper is submitted with the approval of the Director of KEMRI.

**Contributors** JN conceived of the idea for the study supported by ME who obtained the funding for this project. Preparation and conduct of the study were undertaken by JN who also undertook all the interviews, observations and the qualitative analysis with support from ME and DG. CJ provided theoretical support during analysis and write up while ME and DG contributed to the analytical interpretation of the data both in discussion with JN. JN produced the draft manuscript to which all authors contributed to its development. All authors read and approved the final manuscript.

**Funding** This work was supported by a joint Health Systems Research Initiative grant provided by the Department for International Development, UK (DFID), Economic and Social Research Council (ESRC), Medical Research Council (MRC) and Wellcome Trust, grant number MR/M015386/1. ME is supported by a Wellcome Trust Senior Research Fellowship (#207522). The funding sources had no role in the study design, writing of the report and in the decision to submit the manuscript for publication. This paper is published with the permission of the Director of KEMRI.

**Competing interests** JN, CJ, DG and ME received research grants linked to work in Kenya on topics related to this report.

**Patient consent for publication** Not required.

**Ethics approval** Ethical approval was obtained from the Kenya Medical Research Institute Ethical Review Committee (approval number SERU 3366). Written informed consent was obtained from all the participants.

**Provenance and peer review** Not commissioned; externally peer reviewed.

**Data availability statement** All data relevant to the study are included in the article. No additional data is available.

**ORCID iD**
Jacinta Nzinga http://orcid.org/0000-0001-8394-8857

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
