## [Reviewer comments · BMJ Open]

ARTICLE DETAILS

TITLE (PROVISIONAL)	The value of Stakeholder Engagement in improving Newborn Care in Kenya: A Qualitative Description of perspectives and lessons learned
AUTHORS	Nzinga, Jacinta; Jones, Caroline; Gathara, David; English, Mike

VERSION 1 – REVIEW

REVIEWER	Tweheyo, Raymond Makerere University College of Health Sciences, School of Public Health
REVIEW RETURNED	09-Nov-2020

GENERAL COMMENTS	Stakeholder engagement in improving newborn care in Kenya: description, perspectives and lessons learned. Nzinga et al., 2020. General comments: A good contextual description of stakeholder engagement for improving newborn services delivery. It would have been interesting to comment on the level of relative power, authority – and delegated authority among stakeholders, and the roles and interests of the various stakeholders that harnessed or curtailed the engagement process as it evolved – since inherently stakeholder engagement entails the prioritization of alternatives for implementation research. As the HSD-N is used as a case-study for reporting the stakeholder engagement process, describe a little more detail about the involvement of the Newborn care service providers in the project. For example, were they implementing stakeholders, researchers, or both? Were they compensated in any way, or did the project follow some kind of public capacity building approach? This helps to delineate their involvement in the project. In places like between pages 18 and 22, quotes could be shortened without loss of content. Also, it appears unusual to box quotes – do check your journal guidelines, to use a relevant quote style. Specific comments: Pg 4. Line 9 “The Kenyan Pediatric Associated” – should this be, “The Kenyan Pediatric Association”?
---

	Pg 4. Line 10 “To create a Clinical Information Network (CIN)” - It would be important to reference this if it is described in detail elsewhere. Also, do comment on the geographic distribution of this CIN, so as to provide more context to the reader. Pg 4. Line 30. HPSR, and LMIC. You introduced the definitional terms to these earlier, but not the abbreviations. Please introduce abbreviations at first introduction of the term, alternatively, use the whole term in full, to ease reading. Pg 4. Lines 55 – 60. Break into two sentences. The message gets lost within the paragraph. Pg 5. Lines 15 – 18 “powerful professionals, health regulators, health professional bodies, private institutions.....” There is need to provide a descriptor of what you mean by powerful, for instance do you mean powerful for various health system functions such as ‘policy making’, ‘policy implementation’ etc?. Could you also describe in brief how these stakeholders and their positions (power influence) were identified? Pg 5. Line 55 “NBUs” – same comment as above. Pg 6. Table 1 could be formatted better (please check with journal guidelines too, for what might be acceptable). I suggest a smaller font size, with single spacing of text, and content auto-fitted to window and text. Pg 7. Line 14 “SOPs” – same comment as above. Reference 32 & 36 seems incomplete, please check and update.
--	---

REVIEWER	Doherty, Tanya Medical Research Council, Health Systems Research Unit
REVIEW RETURNED	11-Nov-2020

GENERAL COMMENTS	Thank you for the opportunity to review this research article reporting on stakeholder engagement in newborn care in Kenya. I would like to commend the authors for writing up these experiences which don’t often get disseminated but are so critical to understanding the sustainability of health systems improvements. The article is submitted as an original research article yet is not written in this style. I would like to suggest some changes to bring it more in line with an original research submission. Alternatively I feel it may be better suited as an analysis article for BMJ Global Health. The abstract should include the research design. A methods section should come before the study background. I would suggest moving the methods description currently on pages 8-10 earlier to after the introduction and include study setting in the methods section. The description of the HSD-N project and phases could be shortened as the figure does provide a good overview. The boxes with the quotes are not very helpful. I suggest either removing quotes from the boxes into the main text or just having one box per theme with all quotes together. Also please add some descriptive tag to each quote indicating participant type (policy maker, health professional etc, and gender). It would also be helpful to have a figure or table with the four key themes and major sub-themes as an overview for readers.
---

	The results from the document analysis of project feedback reports and longitudinal observations of 20 meetings don't appear to be described in the results, only the individual interviews. Also figure 2 is missing. If these are reported separately they could be removed from the methods description for this paper. For an original research paper the lessons and implications section should be converted into a discussion section. The COREQ checklist has not been correctly completed. The authors should indicate the page numbers where each of the topics are described.
--	---

VERSION 1 – AUTHOR RESPONSE

REVIEWER #1:		
Overview		
A good contextual description of stakeholder engagement for improving newborn services delivery. It would have been interesting to comment on the level of relative power, authority – and delegated authority among stakeholders, and the roles and interests of the various stakeholders that harnessed or curtailed the engagement process as it evolved – since inherently stakeholder engagement entails the prioritization of alternatives for implementation research.	Thank you for this valuable comment. We agree that issues of power analysis and the effect on the stakeholder engagement would have been useful to explore. While as research team we often reflected and discussed these issues throughout the project life, it was beyond the scope of this manuscript to explore them in depth. We have however included this as an area for future research	The need to focus on power analysis is now highlighted on page 19, line 29-30
As the HSD-N is used as a case-study for reporting the stakeholder engagement process, describe a little more detail about the involvement of the Newborn care service providers in the project. For example, were they implementing stakeholders, researchers, or both? Were they compensated in any way, or did the project follow some kind of public capacity building approach? This helps to delineate their involvement in the project.	The newborn care service providers in this study include both the frontline workers and health care managers As highlighted on page 10, their involvement in the HSD-N project is described in detail in Table 2 We have included a note that none of the HSD-N stakeholders were compensated for their involvement. However, we did provide capacity building on research	These changes are highlighted on page 9, 8-10

	methods and select clinical care topics	
In places like between pages 18 and 22, quotes could be shortened without loss of content. Also, it appears unusual to box quotes – do check your journal guidelines, to use a relevant quote style.	We have reduced the length of the quotes for coherence We used box quotes to make sure article does not exceed 4000 words. However, we have reduced the number of box quotes to only 1-2 per theme	These changes are highlighted on pages 13 through to page 18
Specific comments		
Pg 4. Line 9 “The Kenyan Pediatric Associated” – should this be, “The Kenyan Pediatric Association”?	We have revised The Kenyan Pediatric Associated” to the The Kenyan Pediatric Association	This change is highlighted on page 4, line 4
Pg 4. Line 10 “To create a Clinical Information Network (CIN)” - It would be important to reference this if it is described in detail elsewhere. Also, do comment on the geographic distribution of this CIN, so as to provide more context to the reader	We have now added a reference describing the creation of CIN and included more detail on the geographic distribution of CIN	These changes are highlighted on page 4, line 5-7
Pg 4. Line 30. HPSR, and LMIC. You introduced the definitional terms to these earlier, but not the abbreviations. Please introduce abbreviations at first introduction of the term, alternatively, use the whole term in full, to ease reading.	We have now introduced the abbreviations of LMIC and HPSR	These changes are highlighted on page 3, line 5 and page 4 line 18
Pg 4. Lines 55 – 60. Break into two sentences. The message gets lost within the paragraph	The text appearing on page 4 line 55-60 been altered entirely therefore the sentence now reads coherently	The changes are highlighted on page 4, line 55-60
Pg 5. Lines 15 – 18 “powerful professionals, health regulators, health professional bodies, private institutions.....” There is need to provide a descriptor of what you mean by powerful, for instance do you mean powerful for various health system functions such as ‘policy making’, ‘policy implementation’ etc?.	We have now provided clarification and description of what we mean by powerful stakeholders A description of how the stakeholder and their relative positions of power were identified has also been added	These changes are provided on page 5, line 9-10

Could you also describe in brief how these stakeholders and their positions (power influence) were identified?		
Pg 5. Line 55 “NBUs” – same comment as above	The text falling under line 55 on page 5 has been altered therefore the ‘NBUs’ no longer appears	N/A
Pg 6. Table 1 could be formatted better (please check with journal guidelines too, for what might be acceptable). I suggest a smaller font size, with single spacing of text, and content auto-fitted to window and text.	To match the journal guidelines, we have reformatted Table 1 and used font 9 with single spacing and auto fitted to window and text	We now provide a revised Table 1 as suggested
Pg 7. Line 14 “SOPs” – same comment as above.	Based on overall reviewers’ comments, this section has now been deleted	N/A
Reference 32 & 36 seems incomplete, please check and update.	We have now amended reference 32 and 36	This change is highlighted in the references, page 23 line 8-10 and line 23-25
REVIEWER #2:		
Thank you for the opportunity to review this research article reporting on stakeholder engagement in newborn care in Kenya. I would like to commend the authors for writing up these experiences which don’t often get disseminated but are so critical to understanding the sustainability of health systems improvements	Thank you for the interest and encouraging comments about our work	N/A
The article is submitted as an original research article yet is not written in this style. I would like to suggest some changes to bring it more in line with an original research submission. Alternatively, I feel it may be better suited as an analysis article for BMJ Global Health.	We have now revised the structure of the paper to meet the guidelines of the journal’s original research submission Please note that the original submission of this manuscript was to BMJ Global Health who in turn advised that the article is better suited for BMJ Open	We highlighted the following changes;  • Included a methods section in the Abstract • Included a study setting section in the Methods section • Included a Discussion and Conclusion section
The abstract should include the research design.	We have included the research design in the abstract	The change in the abstract is highlighted on page 1, line 26-27

A methods section should come before the study background. I would suggest moving the methods description currently on pages 8-10 earlier to after the introduction and include study setting in the methods section	We appreciate this suggestion on having the methods appear earlier on in the write up. The methods do appear after the Introduction. However, the introduction section remains lengthy owing the detailed explanation of the background of the project which guides the design and methodology of the study. Nonetheless, we have now revised the methods by including a study setting section within it	These changes to the methods section are highlighted on pages 7 through to 8 We have included a study setting section as highlighted on page 7, line 5-11
The description of the HSD-N project and phases could be shortened as the figure does provide a good overview.	We have now reduced the description of the HSD-N project	This change is highlighted on pages 4-5
The boxes with the quotes are not very helpful. I suggest either removing quotes from the boxes into the main text or just having one box per theme with all quotes together. Also please add some descriptive tag to each quote indicating participant type (policy maker, health professional etc, and gender).	We have revised the formatting of the results section so that there are at least only 1-2 boxes per theme with accompanying quotes included therein We have also included descriptive tags for each of the quotes from the various participants	These changes are highlighted throughout the results section on pages 9-19
It would also be helpful to have a figure or table with the four key themes and major sub-themes as an overview for readers.	We have now included a table with all the 4 themes and sub-themes	We have now included Table 2 on page 9, line 4 at beginning of the results section to summarize the themes
The results from the document analysis of project feedback reports and longitudinal observations of 20 meetings don't appear to be described in the results, only the individual interviews. Also figure 2 is missing.	Figure 2 was erroneously included in the submission of this manuscript, but we have now corrected this. While we did not systematically record the findings from the document analysis and longitudinal	These changes are highlighted on page and on page 6, line 1 of the introduction and on page 8 of the methods section line

If these are reported separately they could be removed from the methods description for this paper.	observations; the review process was useful in informing our analysis of the interviews. However, we agree with the suggestion to remove these from the methods description section	
For an original research paper, the lessons and implications section should be converted into a discussion section.	We have now converted the lessons and implications sections into a discussion section	These changes are highlighted on page 19, line 1-32
The COREQ checklist has not been correctly completed. The authors should indicate the page numbers where each of the topics are described	We have now corrected the COREQ checklist	The COREQ checklist now includes page numbers where each of the topics are described

VERSION 2 – REVIEW

REVIEWER	Tweheyo, Raymond Makerere University College of Health Sciences, School of Public Health
REVIEW RETURNED	06-Apr-2021

GENERAL COMMENTS	The authors have satisfactorily responded to my concerns and comments. This version has extensive revisions, is more coherent in terms of aligning to the format of original research articles.
--

REVIEWER	Doherty, Tanya Medical Research Council, Health Systems Research Unit
REVIEW RETURNED	23-Mar-2021

GENERAL COMMENTS	We have now included Table 2 on page 9, line 4 at beginning of the results section to summarize the Themes' – this is missing from the revised paper. There is no study design sub-section in the methods description. The COREQ checklist is still incorrect. There should be no responses in the column 'reported on pg no' simply write the page number where the topic is described.
--

VERSION 2 – AUTHOR RESPONSE

Reviewer: 1

Dr. Raymond Tweheyo, Makerere University College of Health Sciences, The University of Manchester

Comments to the Author:

The authors have satisfactorily responded to my concerns and comments

RESPONSE: Many thanks for your review

Reviewer: 2

Dr. Tanya Doherty, Medical Research Council

Comments to the Author:

'We have now included Table 2 on page 9, line 4 at beginning of the results section to summarize the Themes' – this is missing from the revised paper.

RESPONSE: Table 2 summarizing the themes and sub-themes is highlighted in the revised paper

There is no study design sub-section in the methods description.

RESPONSE: We have now included a study design section on page 7, line 4-5

The COREQ checklist is still incorrect. There should be no responses in the column 'reported on pg no' simply write the page number where the topic is described

RESPONSE: We have now revised the COREQ checklist to only report the page number